# Myostatin as a Biomarker of Muscle Wasting and other Pathologies-State of the Art and Knowledge Gaps

**DOI:** 10.3390/nu12082401

**Published:** 2020-08-11

**Authors:** Jan Baczek, Marta Silkiewicz, Zyta Beata Wojszel

**Affiliations:** 1Department of Geriatrics, Medical University of Bialystok, Doctoral School, Fabryczna 27, 15-471 Bialystok, Poland; 2Faculty of Medicine with the Division of Dentistry and Division of Medical Education in English, Medical University of Bialystok, Kilinskiego 1, 15-089 Bialystok, Poland; 35150@student.umb.edu.pl; 3Department of Geriatrics, Medical University of Bialystok, Fabryczna 27, 15-471 Bialystok, Poland; beata.wojszel@umb.edu.pl; 4Department of Geriatrics, Hospital of the Ministry of Interior and Administration in Bialystok, Fabryczna 27, 15-471 Bialystok, Poland

**Keywords:** sarcopenia, myostatin, muscle wasting, biomarker, geriatrics, older patients

## Abstract

Sarcopenia is a geriatric syndrome with a significant impact on older patients’ quality of life, morbidity and mortality. Despite the new available criteria, its early diagnosis remains difficult, highlighting the necessity of looking for a valid muscle wasting biomarker. Myostatin, a muscle mass negative regulator, is one of the potential candidates. The aim of this work is to point out various factors affecting the potential of myostatin as a biomarker of muscle wasting. Based on the literature review, we can say that recent studies produced conflicting results and revealed a number of potential confounding factors influencing their use in sarcopenia diagnosing. These factors include physiological variables (such as age, sex and physical activity) as well as a variety of disorders (including heart failure, metabolic syndrome, kidney failure and inflammatory diseases) and differences in laboratory measurement methodology. Our conclusion is that although myostatin alone might not prove to be a feasible biomarker, it could become an important part of a recently proposed panel of muscle wasting biomarkers. However, a thorough understanding of the interrelationship of these markers, as well as establishing a valid measurement methodology for myostatin and revising current research data in the light of new criteria of sarcopenia, is needed.

## 1. Introduction

Sarcopenia, as currently defined by the European Working Group on Sarcopenia in Older People 2 (EWGSOP2), is a syndrome comprised of deterioration of muscle strength, quantity, quality and performance [1]. It is a geriatric syndrome occurring with advancing age, with a prevalence (estimated by earlier EWGSOP criteria) of 12.9% (95% confidence interval: 9.9–15.9%) among community-dwelling older adults [2]. With its high prevalence and proved impact on disability and hospitalization risk, quality of life and mortality, sarcopenia poses a major health issue for older people and a public health burden [3,4,5,6]. The pathogenesis of this disorder comprises of various factors and mechanisms [7,8], including genetic variability [9,10], cytokine activity [11,12,13], vitamin D, testosterone and growth hormone deficiency [14,15,16,17], mitochondrial dysfunction and apoptosis [18,19,20], muscle disuse [15,21,22], nutritional status [23,24], atherosclerosis [25,26] and diabetes [27,28]. Some authors distinguish primary (or age-related sarcopenia), occurring in otherwise healthy, usually aged persons, from secondary sarcopenia, the result of chronic inflammatory statuses, diabetes, hormonal alterations, vascular disturbances, renal, respiratory and/or cardiac failure and immobilization [29,30]. Furthermore, sarcopenia remains intertwined and overlapping with other syndromes such as frailty [31,32] and cachexia [33,34], and its development significantly worsens the prognosis of various diseases such as heart failure [35,36], chronic kidney disease [37] and liver failure [38]. Muscle wasting in aging (sarcopenia) shares many common metabolic pathways and mediators with body wasting (cachexia) observed during cancer and other chronic diseases, such as multiple inflammatory organ-specific disease or cardiovascular disease [39]. Both sarcopenia and cachexia are characterized by inflammation and oxidative stress, where specific regulating molecules associated with wasting are either activated or repressed, but they are distinct muscle wasting entities [40]. In cachexia, body wasting involves not only muscle, but also adipose and bone tissue, therefore—apart from muscle loss—a marked weight loss, anorexia, asthenia and anemia are observed. A degree of cachexia—usually reported as a complication of chronic diseases—is inversely correlated with the survival time of the patient, and it always implies a poor prognosis [41]. The Glasgow Prognostic Score (based on the assessment of albumin and C-reactive protein (CRP) levels) or its modified version (including also complete blood count) can be used to distinguish cachexia and secondary sarcopenia [42,43].

Early diagnosis of sarcopenia is crucial for the effectiveness of a possible intervention; however, despite the new algorithm proposed by EWGSOP 2, it remains a difficult task. A feasible biomarker of sarcopenia could become an essential tool facilitating and accelerating the diagnostic process, although an applicable molecule is yet to be established. A valid biomarker should be characterized by high sensitivity, specificity and predictive power [44]; however, finding a substance characterized by such features is difficult given the pluricausal nature of sarcopenia. That is why, as proposed by Calvani et al., creating a panel of complementary biomarkers, instead of looking for a universal one, emerges as a possible solution to this issue [45].

Myostatin, also known as growth differentiation factor -8 (GDF-8), is a chalone, a transforming growth factor β (TGF-β) superfamily member acting as a negative regulator of muscle growth. It was first reported by McPherron et al., who discovered that myostatin gene deletion led to hypermuscularity in mice [46]. Similar effects were demonstrated in cattle, sheep, dogs and humans [47,48,49,50], which made myostatin a point of interest in the agriculture industry and medicine. Myostatin is most abundant in skeletal muscle; however, its expression was noted in cardiac muscle and in adipose tissue [51,52]. It is synthesized as an inactive precursor protein and requires a two-step proteolytic cleavage to reach its mature form. The first cleavage is made by furin family enzymes, which remove the signal peptide producing latent myostatin complex found in the serum. This complex consists of mature myostatin dimerized by disulfide bonds, which remain non-covalently bound with and inhibited by myostatin propeptide. The second cleavage occurs in extracellular space and is made by bone morphogenetic protein 1 (BMP1)/Tolloid matrix metalloproteinases, which separate active myostatin ligand from the inhibitory N-terminal propeptide domain, allowing interaction with the receptor [53,54,55]. Active myostatin dimerized by disulfide bonds binds with transmembrane activin type IIB receptor (ActRIIB), which signals through activin receptor-like kinase 4 or 5 (ALK4 and ALK5) [53]. These kinases phosphorylate SMAD family members 2 and 3 (SMAD2 and SMAD3), which form a complex with SMAD4. This complex translocates into the nucleus and affecting transcriptional factors, such as the myocyte-specific enhancer factor (MEF2), and the myoblast determination protein 1 (MyoD) inhibits myoblast proliferation and differentiation [53,56,57,58]. Furthermore, myostatin modulates Akt pathway activity, effectively inhibiting muscle hypertrophy through the mammalian target of rapamycin (mTOR), and increasing muscle degradation via the forkhead box protein O1 (FoxO1) pathways [59]. Myostatin was also found to signal via the nuclear factor kappa-light-chain-enhancer of activated B cells (NF-κB), which, together with SMAD3 activation, could affect glucose uptake inducing insulin resistance [60,61].

Regarding the regulation of myostatin expression, various mechanisms affect its transcription and translation. One of such mechanism is interaction with myostatin gene promoter. A research on the 5’-terminal regulatory region of the myostatin gene revealed various agents possibly acting as transcriptional factors, such as myocyte enhancing factor 2 (MEF2), peroxisome proliferator-activated receptor gamma (PPARγ), MyoD and NF-κB, as well as a mechanism for myostatin upregulation by glucocorticoids [62]. Myostatin gene expression was also found to be affected by tumor necrosis factor -α (TNF-α), hormones (such as insulin-like growth factor (IGF-1), angiotensin II and thyroid hormones), nutritional supplementation, hyperammonemia, physical exercise and hypoxia [63,64,65,66,67,68,69,70]. Furthermore, myostatin was found to down-regulate its expression in a self-regulatory loop [71]. A novel mechanism regulating myostatin mRNA expression is associated with the activity of microRNAs (miRNA) [48,68,72]. These non-coding RNAs were found to inhibit myostatin mRNA translation, leading to muscle hypertrophy in sheep. Furthermore, miRNA expression was found to be affected by essential amino acids and, indirectly, by myostatin itself, which demonstrates an important myostatin regulation feedback mechanism [68,73]. More recent reports on various factors affecting myostatin mRNA expression can be found in suitable paragraphs below. It is worth mentioning that changes in myostatin mRNA expression do not always correspond with muscle protein, and the serum myostatin level and the results of studies on this relationship seem contradictory [64,69,74,75]. That is why, in this research, we focused on data regarding circulating myostatin concentration, as it affects myostatin potential as a biomarker in a more direct and explicit manner. A more detailed description of myostatin structure, mechanism and regulation can be found in several thorough reviews [76,77,78].

Due to a high abundance in skeletal muscle and its function as a myokine, myostatin is posing as a potential biomarker of muscle wasting and therapeutic opportunity for sarcopenia and cachexia. The aim of this review is to verify myostatin’s potential as a serum biomarker in the light of the latest studies, examine various knowledge gaps impeding its usage in diagnosing muscle wasting and also to discuss new findings on this promising peptide and the opportunities they reveal.

## 2. Myostatin and Muscle Wasting

It has been long known that myostatin acts as myokine, negatively regulating skeletal muscle mass. Although a number of novel research studies have been focused on the relationship between serum myostatin and muscle mass and function, and thus, its potential as a biomarker, the results seem unclear. Most studies indicate that a higher circulating myostatin concentration can be found in patients with higher muscle mass [79,80,81,82,83,84,85], but a few research studies resulted in finding such an association only in men [86,87], and some did not show any correlation with muscle mass parameters at all [88,89,90,91]. A study on healthy, obese and metabolically unhealthy participants found a contradictory, significantly negative correlation between muscle mass and serum myostatin, suggesting a major influence of metabolic syndrome on circulating myostatin level [92]. Similarly, contradictory findings have been made regarding the relationship between serum myostatin and muscle function. Some studies found increased circulating myostatin concentrations in patients with better muscle function or physical performance [80,81,83,84,87], and several indicated such a relationship to be more apparent in the male population [85,93], while others did not show any clear correlation between those parameters [82,88,89,92,94].

There are scarce reliable reference data comparing sarcopenic and robust patients with regard to serum myostatin concentration. In their large cross-sectional study, Han et al., found that serum myostatin level might be treated as a significant risk factor of pre-sarcopenia (defined as low muscle mass regardless of muscle function) [95]. Similarly, Tay et al., found myostatin to be a significant risk factor of sarcopenia, but only in the male population, and identified no significant difference in serum myostatin between sarcopenic and non-sarcopenic patients [96]. Hofman et al., noticed no difference in serum myostatin levels between sarcopenic and healthy elderly women [89]. One study found higher myostatin gene expression in sarcopenic patients; however, circulating myostatin concentrations were not measured [97]. Regarding muscle wasting in cancer cachexia, studies on patients with lung, colorectal and medullary thyroid cancer demonstrated that plasma myostatin concentration was decreased in patients with cancer cachexia compared to non-cachectic patients [81,98]. Furthermore, the serum myostatin level was found to be an independent predictor of overall survival in older patients with solid metastatic tumors [99].

Recent studies on neuromuscular disorders, which result in muscle hypotrophy and wasting, emerge as an example of myostatin used as a feasible biomarker. A study by Burch et al., indicates that decreased serum myostatin levels in muscle pathologies are associated with disease progression [100], which can be explained by the down-regulation of myostatin pathway in order to compensate for muscle wasting or atrophy [101]. Further research present myostatin as a feasible biomarker for disease progression, severity and treatment efficacy in mice and patients with myotubular myopathy [102], as well as for estimating a therapeutic response to pharmaco-gene therapy in Duchenne muscular dystrophy [103]. A similar mechanism emerges in idiopathic inflammatory myopathies. Vernerova et al., demonstrated that such disorders with decreased serum myostatin level associated with an increase of circulating follistatin concentration pose as a possible compensatory mechanism attempting to counteract muscle hypotrophy by inhibiting myostatin/ActRIIB signaling in such disorders [104]. All these novel findings greatly expand the understanding of myostatin in neuromuscular diseases and prove its viability as a biomarker in these conditions. These results suggest that while both neuromuscular diseases and sarcopenia are associated with changes in myostatin abundance, the mechanism responsible for such variations is different. While, in a number of myopathies, myostatin is downregulated in order to counteract muscle wasting and, thus, seems to be a feasible biomarker of these disorders, in sarcopenia, the issue seems more complex. Serum myostatin level in muscle wasting of the elderly poses as a result of myostatin expression regulation and decreased muscle mass, which are influenced by other factors further discussed in this review. Such a variety of aspects affecting circulating myostatin concentration is a possible explanation of the aforementioned data discrepancy.

Despite the fact that such inconsistencies in results might be discouraging for considering serum myostatin as a valid biomarker of sarcopenia, we believe that it is important to research into possible causes of such differences between studies. Even if myostatin does not prove a valid candidate for a singular biomarker, it might become an important part of a panel of markers, as proposed by Calvani et al. [45]. However, before incorporating myostatin into such a panel would be feasible, many obstacles must be overcome, with discrepancies in results being the first one. That is why we would like to discuss possible explanations of inconsistent results regarding serum myostatin levels below.

Primarily, there is a strong need to validate the methods used to measure serum myostatin abundance. There is a significant sequence similarity between myostatin and growth differentiation factor -11 (GDF-11), which caused conflicting results in the past, as some antibodies used in detection methods bind both substances [105,106,107]. Although early issues with the specificity of assay methods seemed to be resolved by the development of enzyme-linked immunosorbent assays (ELISA) [108,109], results of novel research using this method still appear as unclear or even contradictory. A possible explanation of this problem might be the fact that some assays detect both active and latent forms of myostatin [109]. Similar observations led Bergen et al., to develop a new liquid chromatography with a tandem mass spectrometry assay to assess serum myostatin abundance [85]. Their results showed gender-specific serum myostatin changes with age as described below, as well as significantly lower myostatin concentrations in sarcopenic women. In men, on the other hand, myostatin appeared as a muscle mass homeostasis regulator, but not a major factor of sarcopenia. However, Bergen et al., defined sarcopenia using muscle mass deficiency only, which asks future studies to validate their results using the latest EWGSOP criteria for diagnosing sarcopenia. Their results were partially confirmed by the SomaScan aptamer technology measurement of samples from two large cohort studies in which, however, sarcopenia was not assessed [110]. Further research investigated various, specific myostatin measurement methods, such as immunoassays, immunoradiometric sandwich assay, antibody-free assay and liquid chromatography-tandem mass spectrometry assay combined with immunoprecipitation [111,112,113,114]. These new, specific assays could be used to clarify discrepancies in earlier results, as well as serve as a validation tool for commercially available methods.

Another issue lies in the necessity of a better understanding of the role of other myokines and biochemical parameters in muscle mass and function regulation, as myostatin is not the only substance that plays an important role in musculoskeletal growth, development and aging. It is worth emphasizing that myostatin acts as a part of the activin A-myostatin-follistatin system. As we have mentioned above, myostatin binds to the type IIB receptor (ActRIIB), whereas activin is thought to bind with greater affinity to the type IIA activin receptor (ActRIIA), and both are involved in the regulation of muscle mass. Follistatin, a single-chain glycosylated protein, is expressed in various tissues and antagonizes both myostatin and activin A activity, influencing their effects on muscle mass [115,116]. Thanks to that, and through stimulating myoblasts to express MyoD, myogenic factor 5 (Myf5) and myogenin, which are myogenic transcription factors, it enhances muscle differentiation, which is a promising agent for improving skeletal muscle healing after injury and muscle diseases [117]. The synthetic N-terminal domain of follistatin-derived fragment peptide DF-3, found recently, effectively inhibited myostatin injected intramuscularly and significantly increased skeletal muscle mass in mice. It is assumed that DF-3 can act as a platform for the development of muscle enhancement based on myostatin inhibition [118].

It is also crucial to establish the influence of insulin-like growth factor-1, testosterone and peptides such as follistatin–related gene (FLRG) protein, GDF-11 (also known as bone morphogenetic protein 11, BMP-11) and growth and serum protein-1 (GASP-1), on muscle mass, sex and age-related differences in their effects, and their relationship with myostatin before it would be possible to assess the feasibility of the latter as a biomarker. Despite multiple recent investigations, further studies are required to fully understand myostatin modulation mechanisms [85,88,89,119,120,121]. In a study by Kalampouka, plasma from older participants was found to reduce myoblast migration and diameter in vitro, in comparison with plasma from younger individuals, despite no difference being detected in plasma myostatin concentration, which demonstrates the importance of other substances and their role in age-related muscle loss [122]. It is especially viable to establish such correlations considering a multi-parameter panel of sarcopenia biomarkers, as mentioned before.

Furthermore, given this multifactorial muscle condition determination and the potential contribution of various substances to regulating their mass and function, it is important not to base conclusions concerning the myostatin effects on univariate analyses only. It would be reasonable to adopt the approach used by many researchers who, in order to produce reliable results and circumvent the impact of serum myostatin changes with various parameters, adjusted myostatin concentration for total body lean mass (which could clarify prior, contradictory results) [85,120], age [120], CRP [87] or performed logistic regressions with many variables to assess the significance of myostatin as a predictor of low muscle mass [86]. Normalizing myostatin levels for different characteristics of the population subject to the study might provide robust data and a better understanding of this myokine; however, deeper knowledge about the relationship of such parameters with serum myostatin is required. That is why, in a further part of this paper, we would like to review novel findings on serum myostatin and its association with physiological parameters such as age, sex, nutritional status and various pathologies focusing on the influence of such correlations on the feasibility of this myokine as a biomarker of muscle wasting.

## 3. Factors Affecting Myostatin Concentration

### 3.1. Age and Sex Differences

Contrary to earliest research [123], most recent studies show no correlation between serum myostatin and age [81,86,87,88,89,121,122]. A study of intramuscular mRNA and protein expression showed that although myostatin gene expression was higher in older adults, no difference in myostatin protein content was found [74]. Results of three studies indicated that serum myostatin concentration rises with age [79,84,94]. In their cross-sectional study, Bergen et al., used a new mass spectrometry-based assay to show that circulating myostatin, acting as a homeostatic regulator of muscle mass in males, could be found in the highest concentration in young men, and decreased with age, while in women, it increased as a function of age, acting as a possible mediator of sarcopenia [85]. Their results were partially confirmed by two further studies. One, using similar methods and conducted at the same research center, found serum myostatin to be decreasing with age in men, yet no significant correlation with age was noted in women [119]. The second one was an analysis of samples collected in two large cohort studies, and it found a decrease of serum myostatin with age in the general population, with a trend for an increase in the female population and a decrease in male patients when analyzed separately [110]. Subsequent studies gave conflicting results concerning sex differences in circulating myostatin level, as it was found to be higher in men [79,84], women [121] or no difference was found [88,93]. More aforementioned studies found sex differences in correlations of circulating myostatin and muscle mass and function. These findings suggest that more research on factors affecting intracellular myostatin protein abundance as well as other plasma myokines is needed to understand the relationship between aging, myostatin and their gender–specific differences.

### 3.2. Physical Activity

Another factor that has to be taken under consideration when measuring serum myostatin is physical activity of the patients. Results of most recent studies suggest that physical exercise attenuates myostatin gene and protein expression in skeletal muscles directly after training [124,125,126,127,128,129,130,131,132,133,134], while a decrease of physical activity achieved by daily step reduction increased myostatin mRNA expression [135]. In most studies, serum myostatin levels were found to increase acutely after various exercise types (ultramarathon, aerobic, resistance and high-intensity training) [136,137,138,139,140,141], and decrease in the long term as a result of physical training [90,142,143,144,145,146]. Two studies noted an increase in circulating myostatin concentrations as a result of various (strength training, balance training, stretching exercises, walking recommendations) physical exercise over very long periods (6 and 12 months) [80,93]. These results demonstrate that physical training affects serum myostatin levels in various ways depending on the time between training and measurement. The acute myostatin level incline was found to begin immediately after the exercise and return to baseline after approximately 24 h; therefore, it would be reasonable to recommend withdrawal from exercise training for a day before taking blood samples from patients [137]. Further research on mechanisms in which physical exercise affects myostatin expressions are required before it could be used as a valid biomarker.

### 3.3. Nutritional and Metabolic Status

The nutritional status is a noteworthy factor affecting serum myostatin abundance. Recent studies demonstrated that circulating myostatin levels are decreased in girls with anorexia nervosa [147] and increased in obese patients [148]. However, Tsioga et al., found no correlation between body mass and serum myostatin among obese patients [149]. This result could be partly explained by a subsequent study by Carvalho et al., which demonstrated that among obese patients, only metabolically unhealthy ones (defined as presenting ≥3 criteria for metabolic syndrome combined with insulin resistance) are characterized by higher circulating myostatin levels, in comparison to healthy lean individuals [92]. The same study found that myostatin levels are correlated with the number of metabolic syndrome criteria met, insulin levels and insulin sensitivity. Other studies confirmed the association of serum myostatin with dysglycemia; the circulating myostatin concentration was found to be higher in prediabetes and the highest in type 2 diabetes mellitus in comparison with healthy controls [150], and it was correlated with fasting plasma glucose, HOMA-IR (Homeostatic Model Assessment for Insulin Resistance), the area under curve for insulin during the oral glucose tolerance test, insulin sensitivity index and serum immunoreactive insulin levels [148,150,151]. Contradictory results were produced in research by Garcia-Fontata et al., which showed lower blood myostatin concentration in patients with type 2 diabetes [152]. Furthermore, myostatin expression in muscles and adipose tissue was demonstrated to be influenced by dysglycemia and insulin sensitivity [134]. Higher serum myostatin levels were found in pregnant (16–20 gestational week) women, who later developed gestational diabetes mellitus, indicating a possible usage of myostatin as a gestational diabetes biomarker [153]. As mentioned before, earlier research found that myostatin might induce insulin resistance via NF-κB and SMAD3 [60,61], and recent animal studies on myostatin function and mechanism brought insight into its role in adipogenesis, communication between adipose tissue and skeletal muscles as well as neural control over insulin sensitivity [154,155,156], yet many aspects remain unknown, and are awaiting in-depth studies. Nevertheless myostatin was pointed out as a potential candidate contributing to sarcopenic obesity that should be addressed in clinical trials with myostatin-inhibitors [157].

### 3.4. Inflammation and Injuries

It is worth mentioning that a significant negative correlation between myostatin and CRP was reported [81,82,84,87]. In one of those studies, a regression analysis stratified for the baseline CRP was conducted, resulting in a significantly steeper dependence curve between myostatin and hand-grip strength in patients with CRP <10 mg/L than in patients with CRP >10 mg/L, suggesting that inflammatory status impacts the performance of myostatin as a biomarker of muscle strength [87]. In his study, Akerfeldt points out the postsurgical decrease of serum myostatin level in acute phase reaction in patients with elective orthopedic surgery and elective coronary bypass surgery, with no differences between these groups [158]. A prospective study carried out by Wallner et al., focused on the role of myostatin in burn-associated muscle wasting, and showed decreased concentration of serum myostatin in the acute phase of burn injury and a reverse effect in the chronic phase [159]. Due to its potential role in the aforesaid disorder, myostatin could be investigated regarding the prediction of the long-term muscle strength, as well as the therapy improving condition of patients after severe burn injuries. Research by Zhao et al., proved that serum myostatin levels are higher in patients with knee osteoarthritis [160]. A novel finding made by Su et al., demonstrated in vivo that overexpression of myostatin induces robust expression of TNF-α in rheumatoid arthritis synovial fibroblasts, leading to chronic synovitis [161]. This study shows a possible crucial role of myostatin in the pathogenesis of rheumatoid arthritis, and sets a ground for further investigations on the matter of possible myostatin-target therapies of the disorder mentioned. In another research on rheumatoid arthritis in remission, serum myostatin was found to decrease in comparison to heathy controls [162]. The research results described above highlight the influence of injuries and inflammation on serum myostatin levels and the importance of inflammatory parameters when considering myostatin as a biomarker of muscle wasting in patients with inflammatory diseases.

### 3.5. Heart

Myostatin became a point of interest in cardiology as a potential muscle wasting biomarker, since cardiac cachexia is a complication of chronic heart failure with dramatically high mortality rates [163]. Despite early studies on myostatin indicating an increase of its serum concentration in heart failure (HF) [164,165], research by Furihata et al., found lower serum concentration of myostatin along with higher levels of its inhibitor—follistatin—in patients with HF [83]. Such unexpected results might arise from the fact that 70% of tested patients performed exercise training during the study. A subsequent examination produced findings consistent with early data, demonstrating higher myostatin myocardial expression in HF (although only in females) [166]. Another research work by Chen et al., found serum myostatin to be higher in chronic HF patients than in healthy controls, higher in non-survivors than in survivors and higher in patients with moderate and high N-terminal pro B-type natriuretic peptide (NT-proBNP) [167]. The same study demonstrated that high circulating myostatin is a risk factor of chronic HF rehospitalization and an independent predictor of mortality. Myostatin was also found to correlate with troponin I peak in patients with acute myocardial infarction, possibly reflecting the extent of damage to the myocardium [168]. Research on patients referred for phase II cardiac rehabilitation indicated no influence of such procedure on serum myostatin level; however, patients with congestive HF were excluded from this study, highlighting the necessity of examining the effect of cardiac rehabilitation on circulating myostatin levels in HF [169]. As demonstrated before by Lenk et al., induction of myostatin expression in HF is possibly mediated by inflammatory cytokines, such as TNF-α, via a p 38 mitogen-activated kinases (p38MAPK)-dependent pathway and NF-κB [64].

In recent years, a number of studies were conducted on circulating and cardiac myostatin in cardiomyopathies of various origins. Myostatin was found to be upregulated in myocardial cells of patients with dilated cardiomyopathy, but not with ischemic cardiomyopathy [170], and higher in abundance both in myocardium and in serum [171]. Fernandez-Sola et al., demonstrated cardiac myostatin upregulation in hearts with cardiomyopathy of hypertensive, alcoholic, valve and coronary origins [172]. Furthermore, Fernlund et al., discovered no difference regarding circulating myostatin concentration in the serum of young patients with hypertrophic cardiomyopathy in comparison to healthy controls, despite lower myostatin levels in patients at risk of this disorder, which indicates a potential role of myostatin in the development of hypertrophic cardiomyopathy [173].

Ju et al., the found serum myostatin level to be significantly higher in patients who developed cor pulmonale as a result of chronic obstructive pulmonary disease (COPD) in comparison with both COPD patients without diagnosed cor pulmonale as well as with healthy individuals [174]. They also managed to find a significant correlation of serum myostatin with plasma BNP levels and right ventricle dysfunction parameters, such as tricuspid annular plane systolic excursion (TAPSE) and right ventricular myocardial performance index (RVMPI). Subsequent studies confirmed those observations [175,176]. Despite new studies on animals demonstrating an anti-hypertrophic and anti-autophagic effect of myostatin on a heart (mediated by 5’AMP-activated protein kinase (AMPK)/mTOR and miRNA-128/PPARγ/NF-κB pathways) [177], as well as its ability to induce interstitial fibrosis in the myocardium via transforming growth factor beta-activated kinase 1 (TAK1) and p38 pathway [178], more research is needed in order to fully understand its role in the pathophysiology of heart disorders.

### 3.6. Kidneys

There are many premises indicating that renal function may play a potential role in myostatin serum abundance. A significant negative correlation of serum myostatin level with kidney function in chronic kidney disease and in autosomal dominant polycystic disease was noticed, even at early stages of kidney failure [133,179,180]. It seems unclear how chronic hemodialysis (HD) and its method affect serum myostatin levels. Contrary to research by Koyun et al., presenting an elevated serum myostatin level in chronic hemodialysis patients compared to healthy controls [91], two other studies showed no difference between these groups [181,182]. Esposito et al., noticed a significant decrease of plasma myostatin (associated by the authors with improvement of malnutrition status) after online hemodiafiltration (HDF) with no changes after bicarbonate hemodialysis (BHD) [181], while further studies showed a significant and comparable decrease of plasma myostatin levels after HD and HDF, and HDF and BHD, accordingly [84,182]. These discrepancies indicate the need for in-depth studies on this matter. It is worth mentioning here that Delanaye et al., found a significant correlation between myostatin and one-year mortality in patients on hemodialysis, which could be explained by the association of this myokine with muscle status and risk of sarcopenia in those patients [84]. As demonstrated by Wang et al., myostatin level changes in chronic kidney disease could result from the upregulation of inflammatory cytokine, TNFα, through activation of the NF-κB pathway, while myostatin activates the ubiquitin-proteasome and autophagy-lysosome systems via Akt and FoxO3 pathways [183].

### 3.7. Gynecological Disorders

In their research, Carrarelli et al., found that myostatin expression and concentration was significantly higher in adenomyotic tissues than in healthy endometrium [184]. Results of a subsequent study indicated that myostatin and its receptors expression rise significantly in deep infiltrating endometriosis and endometrial adenocarcinoma in comparison to control endometrium, which suggests a potential role of myostatin in the aforementioned disorders [185]. A study on the expression of myostatin in preeclampsia revealed elevated levels of myostatin in the plasma of women who later developed this disorder, which can suggest the potential usage of plasma myostatin as biomarker in preeclampsia [186]. A prospective study by Tsigkou showed that high expression of myostatin mRNA is significantly correlated with the upregulation of matrix metalloproteinase 14 mRNA in uterine leiomyoma and the associated high expression of those two with the intensity of dysmenorrhea in the disease mentioned [187]. This novel finding sets the ground for further research on the role of myostatin in uterine pain perception and indicates the necessity of investigating the impact of myostatin expression in various neoplasms on the circulating myostatin level.

### 3.8. Pharmaceutics

Not only natural physiological substances and pathological processes, but also various drugs can influence the expression and level of myostatin in muscles and serum. Many recent studies suggest the potential impact of treatment with various substances (such as acetaminophen, losartan, insulin, denosumab, dapagliflozin or glucocorticoids) on the serum myostatin level.

Earlier observations indicated that acetaminophen or ibuprofen consumption combined with resistance training promoted muscle mass and strength [188]. COX-inhibiting drugs, such as acetaminophen, impacted the adaptive response of skeletal muscle to resistance exercise through altering the cellular mechanisms and suppressing the early response of the mammalian target of rapamycin complex 1 (mTORC1) activity, but also the subtle alterations in myostatin and myogenic factor 6 (MYF6) expression induced by acetaminophen were observed [189].

It is well known that angiotensin II (AngII) plays a critical role in cardiac remodeling and promotes cardiac myocyte hypertrophy. Myostatin is increased in a hypertrophied and infarcted heart. AngII enhanced myostatin protein and mRNA expression in cultured rat neonatal cardiomyocytes in a time-dependent manner [65]. Blocking the angiotensin II type I receptor (AT1R) with losartan before acute heavy-resistance exercise resulted in greater suppression of myostatin messenger RNA compared to placebo [190]. It is, therefore, expected that long-term intervention with this drug may promote muscle growth.

It was suggested that increased production of myostatin could play a critical role in glucocorticoids-induced muscle atrophy [191]. Although Wang and co-workers’ study revealed that dexamethasone significantly decreased (*p* < 0.05) protein synthesis rates while increasing the abundance of myostatin, they confirmed also that the latter is not the main pathway associated with the suppression of muscle protein synthesis by glucocorticoids. They demonstrated that dexamethasone induced the upregulation of myostatin in a time-dependent manner in an in vitro study [192]. The authors suggested that the myostatin signaling pathway is associated with glucocorticoid-induced muscle protein catabolism at the beginning of exposure only, and it was in line with the previous studies in vivo [193]. It was also observed recently that different synthetic glucocorticoids can lead to different muscle effects affecting secondarily different signaling mechanisms.

It is well known that insulin has anabolic effects on skeletal muscle, but knowledge on the molecular mechanisms underlying this effect and on the myostatin role in humans is still not consistent. In the recent in vitro study, the infusion of insulin during euglycemic-hyperinsulinemic clamp (EHC) raised the expression of myogenic growth factors, myogenin and myogenin differentiation protein, and reduced the expression of muscle hypertrophy suppressor, myogenic regulatory factor 4 (MRF4), whereas the expression of myostatin did not change [194]. Contrary to those findings in the acute insulin intervention, the long-term insulin glargine treatment inhibited activation of myokines (IL-6, IL-15, fibronectin type III domain-containing protein 5 (FNDC5), the precursor of irisin, and myostatin) in rats [195]. Also, other glucose-lowering medications can influence serum myostatin levels and muscle condition. In the study by Yamakage at al., dapagliflozin add-on therapy in patients with type 2 diabetes reduced myostatin levels significantly, and they maintained skeletal muscle mass [196].

Summarizing, it is important to thoroughly assess the influence of different therapies on circulating myostatin concentrations before using it to diagnose muscle wasting in patients receiving the aforementioned medications.

## 4. Conclusions

Recent studies, as reviewed above, show that myostatin deserves the attention it received from the scientific community in the past decade, as it could prove to be a viable biomarker of muscle wasting and a useful predictor of mortality in various conditions. One of its main weaknesses lies in low potential specificity, as a number of different factors affect its serum concentration (Figure 1). Yet, myostatin might become an important part of a muscle wasting biomarker panel that could compensate its weaknesses using other, more specific substances. However, before the incorporation into such a panel could become feasible, further research needs to provide deeper knowledge on the mechanism and modulation of myostatin, a better understanding of the way various conditions affect its serum expression and reliable methods of measuring its abundance in blood. Furthermore, myostatin’s ability to predict sarcopenia has to be revised using the latest EWGSOP2 criteria. This goal requires a high level of effort from the academic community. It could, however, result in a powerful tool of early sarcopenia diagnosis, which, especially in patients with co-morbidities such as heart failure, would enable early intervention and mortality reduction. The road to using myostatin as a biomarker in the diagnosis of sarcopenia and muscle wasting seems long, but it is worth taking.

## Figures and Tables

**Figure 1 nutrients-12-02401-f001:**
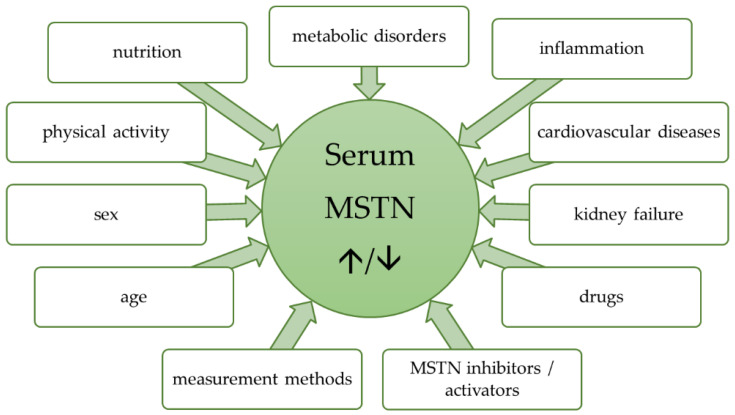
Various factors affecting serum myostatin level.

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
