# Peer review of "Myostatin as a Biomarker of Muscle Wasting and other Pathologies-State of the Art and Knowledge Gaps"

_nutrients, 2020, doi:10.3390/nu12082401_

Round 1
Reviewer 1 Report
It is an interesting review but it is written like a catalogue, and sometimes lacks analysis or reasoning.
A paragraph on the expression of follistatin in relation with sarcopenia is missing.
The authors should also add a paragraph on the regulation of myostatin expression at mRNA level in skeletal/cardiac muscles. In what extend is it linked to protein level and sarcopenia?
P2 line 56: mutations leading to muscle hypertrophy were also observed in sheep and dog
P2 line 63: the mature part of myostatin and the propeptide remains non covalently bound, forming the latent complex, until it interacts with the receptor.
P2 line 63: N-terminal (not n-terminal)
line 117: for the ELiSA: GDF8 and GDF11 show high percentage of similarity and some antibodies may not be able to differentiate the 2 proteins. And also, antibodies may be able to recognize active and/or latent GDF8.
Line 143: it might be reasonable to adjust myostatin concentration for the total body lean mass only if mRNA levels in skeletal muscles are similar in sarcopenic and healthy people, but it was shown that myostatin gene expression was higher in older adults.
3.2: how fast is myostatin regulation after physical activity and how long does it last? What could be the consequences?
The authors should mention 2 major articles recently published by Koch et al (2020) and Mariot et al (2020) showing that myostatin is a circulating biomarker for 2 neuromuscular diseases DNM2 and DMD. The authors should also discussed the differences in myostatin expression in sarcopenia and muscle disorders. Indeed, both neuromuscular diseases and sarcopenia lead to muscle wasting but a in neuromuscular disorders this is associated with a decrease in myostatin synthesis in order to counterbalance the muscle wasting process (Burch 2017; Mariot 2017) whereas in old people, myostatin levels are high, and may act as a mediator of sarcopenia.
Reviewer 2 Report
Baczek and colleagues present a review on myostatin as a biomarker of sacopenia and various factors affecting serum myostatin levels. The review is informative and well organized. It provides a good comprehensive summary of the field.
I believe this review will be more informative if the authors include additional section about “measurement methods” as the authors showed in Figure 1. The mature active region of GDF11 is highly similar to that of myostatin. Multiple studies using different detection methods have reported conflicting age-related changes in GDF11 levels. Egerman et al. found that a previously used GDF11 SOMAmer and GDF11 antibodies bind to both GDF11 and myostatin (Loffredo etal. Cell(2013), Sinha et al. Science(2014), Egerman et al. Cell Metab(2015))
I also believe the review will be improved if the authors could include Tony Wyss-Coray’s reports published in Nat. Med (2019); Undulating changes in human plasma proteome profiles across the lifespan. They revealed that myostatin is decreased with age in mice and human.
Reviewer 3 Report
The authors present a well thought out review of myostatin alterations in several muscle wasting pathologies. They do a good job of identifying different factors that could affect the concentrations of myostatin.
There were a few areas that the authors should address/modify.
1) The authors need to be more consistent in the use of the term sarcopenia, cachexia and muscle wasting. Sarcopenia and cahcexia are not the same thing and the authors should make sure this is clear to the readers. The introduction is almost entirely focused on sarcopenia.
2)The subtopics are arranges well; however the authors provide superficial information about many of the topics. It would be nice to see them discuss some of the mechanisms in which these subtopics affect myostatin not just that they do affect myostatin. They take the time to explain how myostatin functions and signals in the introduction and then no not mention how any of the subtopics can affect these pathways.
3)In section 3.2 the authors should provide more detail on the types of exercise that were used in the studies that they reference. Resistance and aerobic exercise have very different adaptations in the muscle which can affect myostatin levels. There is reserch done on how exercise affects myostatin expression in skeletal muscle and adipose tissue.
4) Section 3.8 This section is very vauge and simply lists some drugs that may alter myostatin levels. There is no indication of how or which direction myostatin levels are altered with these drugs. The authors should consider elaborating this section or removing it as in its current state it does not add much to the paper. Myostatin inhibitors/activators are listed on figure 1 but not discussed in the paper. These could be added to section 3.8.
